# Enteroendocrine cell lineages that differentially control feeding and gut motility

**Marito Hayashi[1], Judith A Kaye[1], Ella R Douglas[1], Narendra R Joshi[1], Fiona M Gribble[2], Frank Reimann[2], Stephen D Liberles[1]\***

[1]Department of Cell Biology, Howard Hughes Medical Institute, Harvard Medical School, Boston, United States; [2]Wellcome Trust MRC Institute of Metabolic Science, University of Cambridge, Cambridge, United Kingdom

**Abstract** Enteroendocrine cells are specialized sensory cells of the gut-brain axis that are sparsely distributed along the intestinal epithelium. The functions of enteroendocrine cells have classically been inferred by the gut hormones they release. However, individual enteroendocrine cells typically produce multiple, sometimes apparently opposing, gut hormones in combination, and some gut hormones are also produced elsewhere in the body. Here, we developed approaches involving intersectional genetics to enable selective access to enteroendocrine cells in vivo in mice. We targeted FlpO expression to the endogenous *Villin1* locus (in *Vil1-p2a-FlpO* knock-in mice) to restrict reporter expression to intestinal epithelium. Combined use of Cre and Flp alleles effectively targeted major transcriptome-defined enteroendocrine cell lineages that produce serotonin, glucagon-like peptide 1, cholecystokinin, somatostatin, or glucose-dependent insulinotropic polypeptide. Chemogenetic activation of different enteroendocrine cell types variably impacted feeding behavior and gut motility. Defining the physiological roles of different enteroendocrine cell types provides an essential framework for understanding sensory biology of the intestine.

**\*For correspondence:**
Stephen_Liberles@hms.harvard.edu

## Editor's evaluation

As digested food moves through the intestines specialized epithelial cells (called Enterochromaffin Cells or EECs) sense and respond to the constituent chemicals. The current work utilizes single-cell transcriptomic analyses and intersectional approaches to define and genetically manipulate subsets of EECs. Key findings are that direct stimulation of EEC subtypes influences key aspects of feeding, specifically gut transit, ingestion, and food preference.

## Introduction

The gut-brain axis plays a critical role in animal physiology and behavior. Sensory pathways from the gut relay information about ingested nutrients, meal-induced tissue distension, osmolarity changes in the intestinal lumen, and cellular damage from toxins (*Bai et al., 2019*; *Brookes et al., 2013*; *Prescott and Liberles, 2022*; *Richards et al., 2021*; *Williams et al., 2016*). Responding neural circuits evoke sensations like satiety and nausea, coordinate digestion across organs, shift systemic metabolism and energy utilization, and provide positive and negative reinforcement signals that guide future consumption of safe, energy-rich foods (*Andermann and Lowell, 2017*; *Sternson and Eiselt, 2017*; *Zimmerman and Knight, 2020*). Moreover, manipulations of the gut-brain axis have been harnessed clinically through gut hormone receptor agonism or bariatric surgery to provide powerful therapeutic approaches for obesity and diabetes intervention (*Richards et al., 2021*; *Seeley et al., 2015*).

Enteroendocrine cells are first-order chemosensory cells of the gut-brain axis and are sparsely distributed along the gastrointestinal tract (*Gribble and Reimann, 2019*). Like taste cells, enteroendocrine cells are epithelial cells with neuron-like features, as they are electrically excitable, release vesicles upon elevation of intracellular calcium, and form synaptic connections with second-order neurons through specialized extrusions called neuropods (*Bohórquez et al., 2015*; *Reimann et al., 2012*). Single-cell RNA sequencing approaches revealed a diversity of enteroendocrine cell types that produce different gut hormones (*Beumer et al., 2018*; *Gehart et al., 2019*; *Haber et al., 2017*). Superimposing cell birthdate on the enteroendocrine cell atlas through an elegant genetically encoded fluorescent clock revealed five major enteroendocrine cell lineages defined by expression of either glucose-dependent insulinotropic polypeptide (GIP), ghrelin, serotonin (called enterochromaffin cells), somatostatin, or a combination of glucagon-like peptide 1 (GLP1), cholecystokinin (CCK), and/or neurotensin (*Gehart et al., 2019*).

Enteroendocrine cell-derived gut hormones evoke a variety of physiological effects (*Drucker, 2016*). GLP1 and CCK are satiety hormones released following nutrient intake, ghrelin is an appetite-promoting hormone whose release is suppressed by nutrients, and serotonin can be released by non-nutritive signals like irritants, force, and catecholamines. Sugar-induced release of GIP and GLP1 causes the incretin effect which rapidly promotes insulin release and lowers blood glucose (*Holst et al., 2009*). CCK, serotonin, and other gut hormones additionally regulate a variety of digestive functions, including gut motility, gastric emptying, gastric acidification, absorption, gallbladder contraction, and exocrine pancreas secretion.

The functions of individual enteroendocrine cell types could in some cases be inferred by summing the actions of their expressed hormones. For example, chemogenetic activation of enteroendocrine cells in the distal colon which express insulin-like peptide 5 triggers a multipronged physiological response that includes appetite suppression through a peptide YY (PYY) receptor, improved glucose tolerance through GLP1, and defecation indirectly through the serotonin receptor HTR3A (*Lewis et al., 2020*). However, a challenge in generalizing this approach is that some enteroendocrine cells release hormones with apparently opposing functions (*Gehart et al., 2019*; *Haber et al., 2017*), and moreover, many gut hormones are also produced by other cell types in the body (*Lee and Soltesz, 2011*; *Okaty et al., 2019*). To overcome these challenges, we developed approaches involving intersectional genetics to obtain highly selective access to major transcriptome-defined enteroendocrine cell lineages. Chemogenetic activation of each of these enteroendocrine cell types produced variable effects on gut physiology and behavior. Obtaining a holistic model for enteroendocrine cell function provides a critical framework for understanding the neuronal and cellular logic underlying gut-brain communication.

## Results and discussion

### Selective access to enteroendocrine cells in vivo through intersectional genetics

We first sought to identify genetic tools that broadly and selectively mark enteroendocrine cells. Transcription factors such as Atoh1, Neurogenin3, and NeuroD1 are expressed in enteroendocrine cell progenitors and/or precursors and act in early stages of enteroendocrine cell development (*Li et al., 2011*). We obtained *Atoh1-Cre* (both knock-in and transgenic lines), *Neurog3-Cre*, and *Neurod1-Cre* mice and crossed them to mice containing a Cre-dependent tdTomato reporter (*Rosa26*$^{CAG-lsl-tdTomato}$ herein defined as *lsl-tdTomato*). *Neurog3-Cre* and *Neurod1-Cre* lines labeled a sparse population of intestinal epithelial cells characteristic of enteroendocrine cells, although the *Neurog3-Cre* line additionally labeled other cells in intestinal crypts and in occasional mice produced broad labeling of intestinal epithelium; neither *Atoh1-Cre* line tested displayed selective labeling of enteroendocrine cells (*Figure 1—figure supplement 1A*; *Schonhoff et al., 2004*). Two-color analysis of tdTomato and gut hormone expression verified tdTomato localization in enteroendocrine cells of *Neurod1-Cre*; *lsl-tdTomato* mice, consistent with prior findings (*Figure 1—figure supplement 1B*; *Li et al., 2012*). Single-cell RNA sequencing of tdTomato-positive cells obtained from these mice (see below) also verified selective enteroendocrine cell labeling.

*Neurod1-Cre* mice provide broad, indelible, and selective marking of enteroendocrine cells within the intestine, but NeuroD1 is also expressed in a variety of other tissues, including the brain, retina,

pancreas, peripheral neurons, and enteric neurons (*Figure 1B and C*, *Figure 1—figure supplement 1D and E*; *Cho and Tsai, 2004*; *Li et al., 2011*). Knockout of NeuroD1 is lethal, causing severe deficits in neuron birth and survival, as well as in the development of pancreatic islets and enteroendocrine cells (*Gao et al., 2009*; *Naya et al., 1997*). We employed an intersectional genetic strategy of combining Cre and Flp recombinases to limit effector gene expression to enteroendocrine cells. *Villin1* (*Vil1*) is expressed with high selectivity in the lower gastrointestinal tract (*el Marjou et al., 2004*; *Maunoury et al., 1992*), so we generated a knock-in mouse allele (*Vil1-p2a-FlpO*) that drives FlpO recombinase expression from the endogenous *Vil1* locus. *Vil1-p2a-FlpO* mice displayed expression of a Flp-dependent *Gfp* allele in epithelial cells throughout the entire length of the intestine with striking specificity (*Figure 1A*, *Figure 1—figure supplement 1C*). Reporter expression was not observed in most other tissues examined, including most brain regions, spinal cord, peripheral ganglia, and enteric neurons; rare GFP-expressing cells were noted in taste papillae, epiglottis, pancreas, liver, and thalamus (*Figure 1C*, *Figure 1—figure supplement 1C and D*; *Höfer and Drenckhahn, 1999*; *Madison et al., 2002*; *Rutlin et al., 2020*). Combining *Neurod1-Cre* and *Vil1-p2a-FlpO* alleles (*Neurod1*[INTER]) yielded highly selective expression of an intersectional reporter gene encoding tdTomato (*Rosa26*[CAG-lsl-fsf-tdTomato] herein defined as *inter-tdTomato*) in enteroendocrine cells, with only occasional cells observed in pancreas, and no detectable expression in other cell types labeled by either allele alone (*Figure 1C*, *Figure 1—figure supplement 1D and E*).

## Charting enteroendocrine cell diversity and gene expression

Our general goal was to use intersectional genetics to access subtypes of enteroendocrine cells that express different gut hormones. We first used single-cell RNA sequencing approaches to measure the extent of enteroendocrine cell diversity, compare findings with existing enteroendocrine cell atlases, and establish a foundation for genetic experiments. Enteroendocrine cells represent <1% of gut epithelial cells, so we used genetic markers for enrichment. NeuroD1 is expressed early in the enteroendocrine cell lineage, and we observed by two-color expression analysis that *Neurod1-Cre* mice target at least several enteroendocrine cell types (*Figure 1—figure supplement 1B*). Since prior enteroendocrine cell atlases were derived from cells expressing an earlier developmental marker, *Neurog3* (*Gehart et al., 2019*), we sought to compare the repertoire of enteroendocrine cells captured by *Neurod1-Cre* and *Neurog3-Cre* mice.

tdTomato-positive cells were separately obtained from the intestines (duodenum to ileum) of *Neurod1-Cre; lsl-tdTomato* mice and *Neurog3-Cre; lsl-tdTomato* mice by fluorescence-activated cell sorting (*Figure 2—figure supplement 1A*). Using the 10X Genomics platform, mRNA was captured from individual cells, and barcoded single-cell cDNA was generated. Single-cell cDNA was then sequenced and unsupervised clustering analysis was performed using the Seurat pipeline (*Hafemeister and Satija, 2019*; *Stuart et al., 2019*). Transcriptome data was obtained for 5,856 tdTomato-positive cells from *Neurog3-Cre; lsl-tdTomato* mice and 1841 tdTomato-positive cells from *Neurod1-Cre; lsl-tdTomato* mice. Twenty-five percent of *Neurog3*-lineage cells (1454/5856) and 87% of NeuroD1-lineage cells (1595/1841) expressed classical markers for enteroendocrine cells (*Figure 2—figure supplement 1B and C*). Moreover, the full diversity of known enteroendocrine cell types was similarly captured by both Cre lines, with *Neurog3-Cre* mice additionally labeling many other cells, including paneth cells, goblet cells, enterocytes, and progenitors (*Figure 2—figure supplement 1C*). These findings are consistent with NeuroD1 acting later than Neurogenin3 in the enteroendocrine cell lineage, but prior to cell fate decisions leading to enteroendocrine cell specialization (*Jenny et al., 2002*).

Since *Neurog3-Cre* and *Neurod1-Cre* mice similarly labeled all known enteroendocrine cell lineages, transcriptome data was computationally integrated for analysis of enteroendocrine cell subtypes. Selective clustering analysis of 3049 enteroendocrine cells from both mouse lines revealed 10 distinct cell clusters, with one cluster representing putative progenitors (*Figure 2A*, *Figure 2—source data 1*). Cell clusters were compared with previously described enteroendocrine cell types based on expression of signature genes encoding hormones and transcriptional regulators (*Figure 2A–C*; *Gehart et al., 2019*). We observed three classes of enterochromaffin cells that similarly express serotonin biosynthesis enzymes (*Tph1*) and associated transcription factors (*Lmx1a*), but differentially produce *Tac1*, *Cartpt*, *Pyy*, *Ucn3*, and *Gad2* (*Figure 2B*). Six other cell types preferentially express either *Gip* (K cells), *Cck* (I cells), *Gcg* (GLP1 precursor, L cells), *Nts* (N cells), *Sst* (D cells), and *Ghrl* (X cells), with L, I,

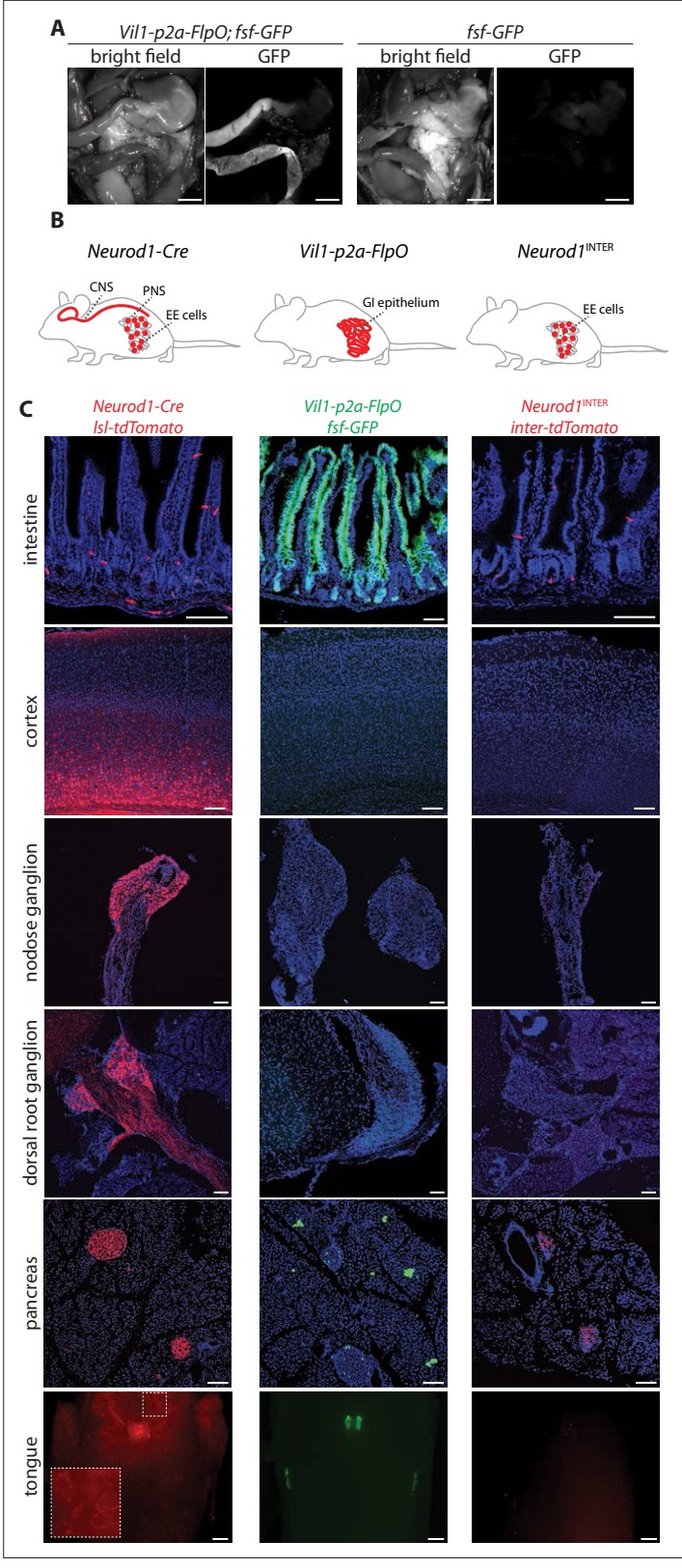

**Figure 1.** Establishing intersectional tools for genetic access to enteroendocrine cells in vivo. (**A**) Bright-field microscopy and native GFP fluorescence microscopy of intestinal tissue from *Vil1-p2a-FlpO; fsf-Gfp* mice (left) and *fsf-Gfp* mice (right). Scale bars: 5 mm. (**B**) Cartoon depicting intersectional genetic strategy to access enteroendocrine cells. (**C**) Native reporter fluorescence in cryosections (20 μm, except 50 μm for cortex and dorsal

*Figure 1 continued on next page*

*Figure 1 continued*

root ganglion) or wholemounts (tongue) of fixed tissues indicated from *Neurod1-Cre; lsl-tdTomato* mice (left), *Vil1-p2a-FlpO; fsf-Gfp* mice (middle), and *Neurod1*INTER*; inter-tdTomato* mice (right). Scale bars: 100 μm for all except 500 μm for tongue. Intestine sections from duodenum (middle) or jejunum (left, right). See *Figure 1—figure supplement 1*.

The online version of this article includes the following figure supplement(s) for figure 1:

**Figure supplement 1.** Characterization of mouse lines for intersectional genetics.

and N cells thought to be derived from a common cell lineage (*Beumer et al., 2020*; *Gehart et al., 2019*). Strong segregation was observed for some signature genes, such as *Tph1* in enterochromaffin cells and *Sst* in D cells. In other cases, signature hormone genes like *Cck* and *Ghrl* were enriched in particular cell clusters but expression was not absolutely restricted and also observed at lower levels in other cell clusters (*Figure 2B*). We note that glutamate transporters were not readily detected in

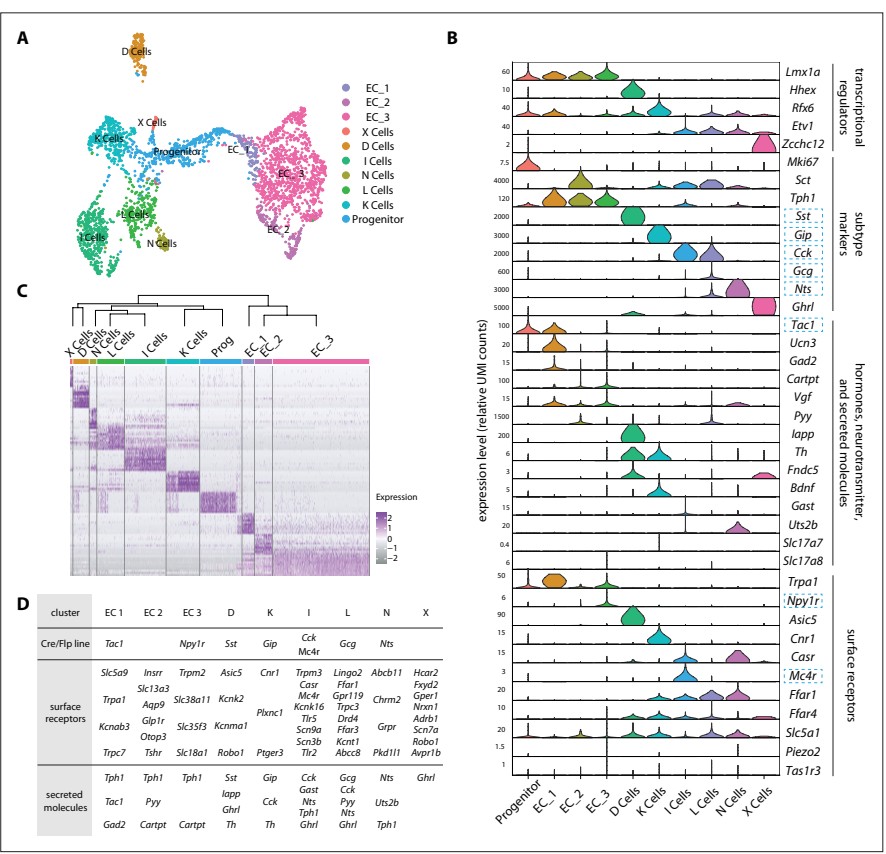

**Figure 2.** An enteroendocrine cell atlas reveals differential hormone and receptor expression. (**A**) A uniform manifold approximation and projection (UMAP) plot of enteroendocrine cell transcriptomic data reveals 10 cell clusters. (**B**) Violin plots showing expression of genes encoding key transcriptional regulators, hormones, other secreted molecules, and receptors across enteroendocrine cell subtypes. Gene loci used for genetic targeting are highlighted with dashed boxes. (**C**) Normalized expression of enriched signature genes (see *Figure 2—source data 1* for a gene list) in single enteroendocrine cells. The dendrogram (top) depicts the relatedness (quantified by position along the Y-axis) between cell clusters based on gene expression. (**D**) For each enteroendocrine cell type, examples of gene loci used for genetic targeting (top, also highlighted in **B**), expressed cell surface receptor genes (middle) and expressed hormone and neurotransmitter-related genes (bottom). Genes were selected among the top 30 differentially expressed genes. See *Figure 2—figure supplement 1*.

The online version of this article includes the following source data and figure supplement(s) for figure 2:

**Source data 1.** Signature genes with differential expression across enteroendocrine cell types.

**Figure supplement 1.** Analyzing cell types marked in different Cre-defined lineages.

**Figure supplement 2.** Analyzing reporter expression in intestine of *Vglut-Cre* mice.

our transcriptomic data (*Figure 2B*, *Figure 2—figure supplement 2*). Thus, each enteroendocrine cell subtype expresses a hormone repertoire with distinct patterns of enrichment but also sometimes partial overlap.

Enteroendocrine cells also express various cell surface receptors to detect nutrients, toxins, and other stimuli. For example, enteroendocrine cells detect sugars through the sodium-glucose cotransporter SGLT1 (encoded by the gene *Slc5a1*), with sodium co-transport thought to lead directly to cell depolarization (*Gorboulev et al., 2012*; *Reimann et al., 2008*). This mechanism is distinct from sugar detection by taste cells or pancreatic beta cells. Gustatory sensations of sweet (and savory/umami) involve taste cell-mediated detection of sugars (and amino acids) through heterodimeric G protein-coupled receptors termed T1Rs (*Yarmolinsky et al., 2009*), while pancreatic beta cells respond to sugar through increased metabolic flux, ATP-gated potassium channel closure, and depolarization. Expression of *Slc5a1* was observed in multiple enteroendocrine cell subtypes, and highest in K, L, D, and N cells, while abundant expression of T1Rs was not detected in any enteroendocrine cell type (*Figure 2B*). These findings are consistent with the ability of taste blind mice lacking T1Rs to develop a preference for sugar-rich foods through SGLT1-mediated post-ingestive signals of the gut-brain axis (*Sclafani et al., 2016*; *Tan et al., 2020*). In addition, free fatty acid receptor genes *Ffar1* and *Ffar4* were broadly expressed in several enteroendocrine cell lineages, but largely excluded from enterochromaffin cells (*Figure 2B*). Orthogonally, the toxin receptor gene *Trpa1* was enriched in enterochromaffin cells (*Bellono et al., 2017*), but not abundantly expressed in other enteroendocrine cells (*Figure 2B and D*). Enterochromaffin cells also reportedly sense force through the mechanosensory ion channel PIEZO2 (*Alcaino et al., 2018*); *Piezo2* transcript was not readily detected in our transcriptomic data, but is enriched in enteroendocrine cells from colon that we did not analyze (*Billing et al., 2019*; *Treichel et al., 2022*; *Figure 2B*). Thus, enteroendocrine cells often express multiple cell surface receptors, suggesting polymodal response properties, and some receptors are expressed by multiple enteroendocrine cell types.

## Genetic access to subtypes of enteroendocrine cells

Next, we obtained genetic tools for selective access to each major enteroendocrine cell lineage. We chose several combinations of Cre and FlpO lines to achieve intersectional genetic access to different enteroendocrine cells based on the cell atlas. (1) *Vil1-Cre; Pet1-FlpE* (*Pet1*^INTER) mice broadly target enterochromaffin cells, while (2) *Tac1-ires2-Cre; Vil1-p2a-FlpO* (*Tac1*^INTER) and (3) *Npy1r-Cre; Vil1-p2a-FlpO* (*Npy1r*^INTER) mice target different enterochromaffin cell subtypes. (4) *Vil1-Cre; Sst-ires-FlpO*, (5) *Gip-Cre; Vil1-p2a-FlpO*, (6) *Cck-ires-Cre; Vil1-p2a-FlpO*, and (7) *Gcg-Cre; Vil1-p2a-FlpO* mice respectively target D, K, I, and L cells (*Figure 2D*), and are herein referred to as *Sst*^INTER, *Gip*^INTER, *Cck*^INTER, and *Gcg*^INTER mice.

Mice of each intersectional allele combination were crossed to *inter-tdTomato* mice, and reporter expression was analyzed across tissues, including in the brain, tongue, airways, pancreas, stomach, and intestine (duodenum to rectum) (*Figure 3—figure supplements 1 and 2*). Each of these seven intersectional combinations produced sparse labeling of intestinal epithelial cells, as expected for labeling of enteroendocrine cell subtypes (*Figure 3—figure supplement 1*). Striking selectivity for enteroendocrine cells was observed across analyzed tissues for intersectional combinations targeting D, K, L, and I cells; sparse labeling was rarely observed in gastric endocrine cells and pancreatic islets, and absent from all other tissues examined. For example, *Cck-ires-Cre* alone (without intersectional genetics) drove reporter (*lsl-tdTomato*) expression in many tissues, including the brain, spinal cord, and muscle, and within the intestine, in enteroendocrine cells as well as enteric neurons, extrinsic neurons, and cells in the lamina propria; however, in *Cck*^INTER; *inter-tdTomato* mice, expression was not observed in the brain, spinal cord, or muscle, and within the intestine, was highly restricted to a subset of enteroendocrine cells, and not observed in other intestinal cell types (*Figure 3—figure supplement 1*). Similarly restrictive reporter expression was observed in *Sst*^INTER; *inter-tdTomato*, *Gip*^INTER; *inter-tdTomato*, and *Gcg*^INTER; *inter-tdTomato* mice. We did note that *Tac1*^INTER and *Npy1r*^INTER alleles more broadly labeled rectal epithelium, and *Npy1r*^INTER additionally labeled taste cells as well as rare cells in the airways and epiglottis (*Figure 3—figure supplements 2 and 3B*). We also note that other genetic tools were inefficient at targeting enteroendocrine cells, including *Nts-ires-Cre* and *Mc4r-t2a-Cre* mice (*Figure 3—figure supplement 3A*).

Hormone expression can be dynamic in individual enteroendocrine cells, and Cre/Flp lines provide an indelible marker for transiently expressed genes (*Beumer et al., 2018*; *Gehart et al., 2019*). Thus, Cre/Flp lines enable in vivo lineage tracing to measure enteroendocrine cell dynamics. We used two-color expression analysis to investigate the repertoires of enteroendocrine cells captured by different intersectional lines. Two-color analysis involved visualization of native reporter fluorescence and immunohistochemistry for GLP1, CCK, SST, and/or serotonin in the duodenum, jejunum, ileum, colon, and rectum (*Figure 3—figure supplement 4*, *Figure 3—figure supplement 4—source data 1*). *Sst*[INTER] mice showed enriched targeting of somatostatin cells throughout the intestine (*Sst*[INTER] cells in duodenum, jejunum, ileum, colon, and rectum: 98.9, 66.4, 66.0, 82.4, and 66.0% express somatostatin, 0.3, 0.0, 0.0, 0.8, and 0.0% express serotonin, 0, 0, 0, 0, and 0% express CCK, and 0, 0, 0.9, 2.3, 0% express GLP1). The *Pet*[INTER] driver also captured cells with other hormones, suggesting that some enteroendocrine cells can either transiently or durably express markers of multiple lineages or can switch identity from enterochromaffin cells to other enteroendocrine cell types (*Pet*[INTER] cells in duodenum, jejunum, ileum, colon, and rectum: 6.3, 2.0, 4.6, 0.9, and 0% express somatostatin, 86.7, 47.9, 43.7, 52.5, and 49.4% express serotonin, 8.2, 6.5, 3.7, 0.3, and 0.6% express CCK, and 4.6, 17.2, 45.7, 18.5, and 25.7% express GLP1). *Tac1-ires2-Cre* and *Npy1r-Cre* both labeled subsets of serotonin cells (100% of labeled cells produce serotonin in each line), with *Tac1-ires2-Cre* labeling a higher percentage of serotonin cells in duodenum (78.4%) than *Npy1r-Cre* (5.0%) (*Figure 3—figure supplement 3A*). Both *Gcg*[INTER] and *Cck*[INTER] mice labeled the majority of GLP1 and CCK cells; these cell types are within the same developmental lineage, and CCK and proglucagon are frequently coexpressed in the same EE cells (*Habib et al., 2012*). *Gcg*[INTER] mice did not effectively label either somatostatin or serotonin cells (*Gcg*[INTER] labeled in duodenum, jejunum, ileum, colon, and rectum 75.5, 67.6, 89.4, 95.9, and 99.0% of GLP1 cells, 34.8, 50.0, 0, 0, and 0% of CCK cells, 0.0, 0.0, 20.8, 34.3, and 22.2% of somatostatin cells, and 1.5, 0.2, 1.8, 0, and 0.8% of serotonin cells). *Cck*[INTER] mice were less selective (*Cck*[INTER] labeled in duodenum, jejunum, ileum, colon, and rectum 68.6, 54.0, 60.7, 71.3, and 23.8% of GLP1 cells, 90.8, 87.9, 96.2, 54.2, and 26.8% of CCK cells, 24.2, 15.1, 28.3, 38.1, and 22.1% of somatostatin cells, and 11.6, 20.6, 15.9, 0.9, and 0.5% of serotonin cells), and a substantial fraction (at least 13.7% in duodenum) targeted other enteroendocrine cells that do not express these four hormones (*Figure 3—figure supplement 4B and C*). It is possible that the *Cck-ires-Cre* allele simply displays inefficient targeting efficiency and/or that it drives reporter expression at early developmental time points with subsequent switching or refinement of cell identity. Together, these experiments measure the extent of selectivity achievable with each genetic tool, with some intersectional combinations providing highly selective genetic access to classes of enteroendocrine cells in vivo.

Next, we assessed the spatial distribution of each enteroendocrine cell lineage along the proximal-distal axis in the duodenum, jejunum, ileum, colon, and rectum by quantifying the number of reporter-positive cells (*Figure 3*). *Pet*[INTER] and *Sst*[INTER] cells were most enriched in the duodenum and colon (*Figure 3*). *Sst*[INTER] cells were the sparsest of enteroendocrine cell types, consistent with observations from scRNA-seq data (*Figures 2A and 3*). *Gip*[INTER] cells and *Gcg*[INTER] cells displayed strikingly distinct spatial patterns. *Gip*[INTER] cells were enriched proximally, with almost no tdTomato+ cells observed in distal intestine. In contrast, *Gcg*[INTER] cells were present along the entire proximal-distal axis and were enriched in colon and rectum. Thus, various enteroendocrine cell subtypes display distinct spatial distributions along the gastrointestinal tract.

## Physiological responses to enteroendocrine cell activation

Direct study of enteroendocrine cell function has been challenging due to a lack of specific genetic tools. Hints come from Neurogenin3 point mutations in human infants or intestine-targeted *Neurog3* knockout, which cause loss of enteroendocrine cells, severe malabsorptive diarrhea, and increased mortality (*Mellitzer et al., 2010*; *Wang et al., 2006*). We sought to develop cell type-specific genetic tools for enteroendocrine cell manipulation, reasoning that they might provide a specific approach to define the repertoire of evoked physiological and behavioral responses.

We first developed chemogenetic approaches for acute stimulation of all enteroendocrine cell types in freely behaving mice. Chemogenetic strategies involved designer G protein-coupled receptors (so-called DREADDs) that respond to the synthetic ligand clozapine-N-oxide (CNO) (*Roth, 2016*). *Neurod1*[INTER] mice were crossed to contain an intersectional reporter allele (*Rosa26*[CAG-fsf-eGFP-FLEX-hM3Dq-mCherry] herein defined as *inter-hM3Dq-mCherry*) that enables expression of a $G\alpha_q$-coupled DREADD

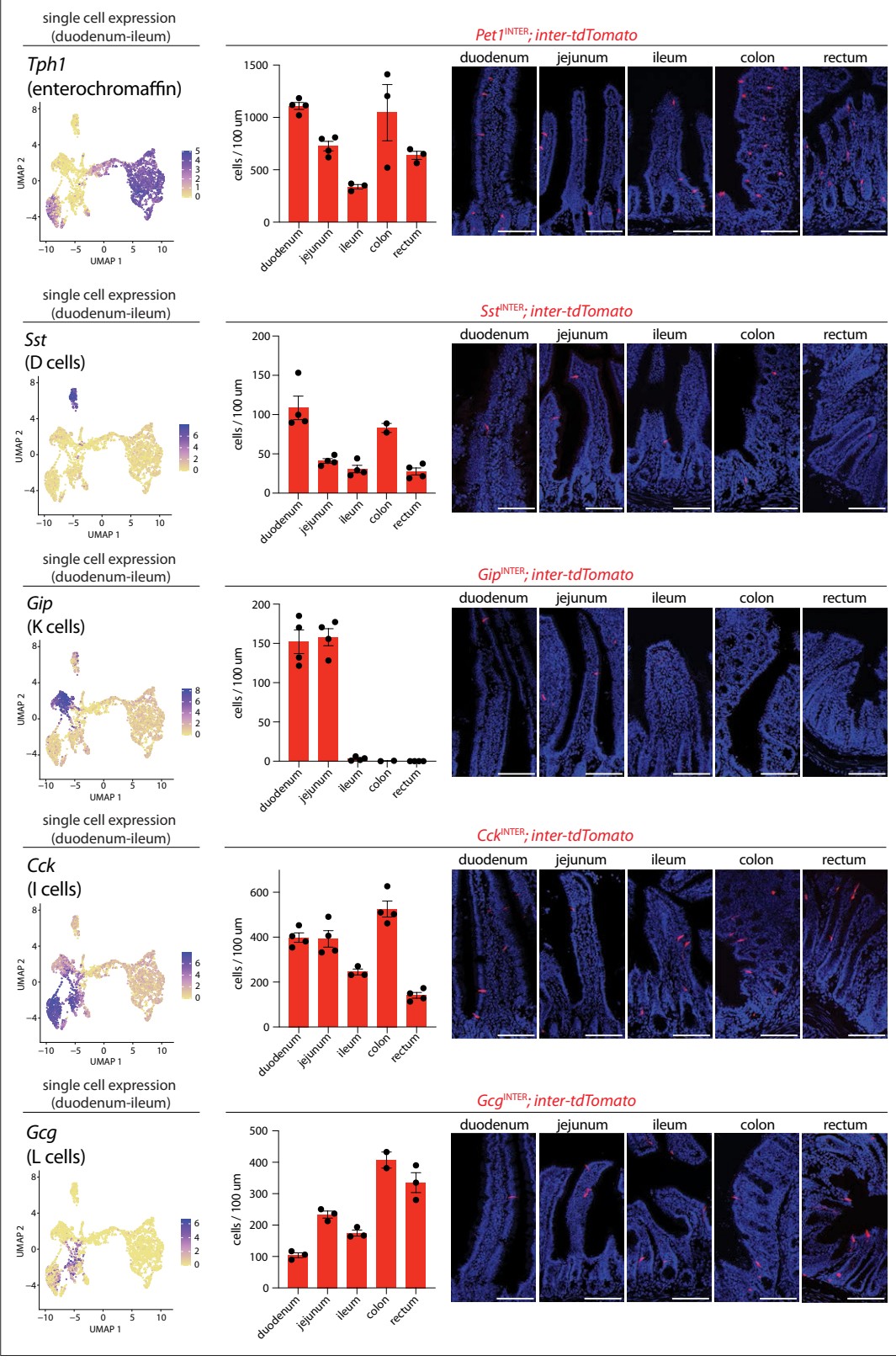

**Figure 3.** Differential targeting of enteroendocrine cell types using intersectional genetic tools. (Left) UMAP plots based on single-cell transcriptome data showing expression of indicated genes across the enteroendocrine cell atlas. (Middle) Number of cells expressing *inter-tdTomato* reporter in five 20 μm sections from intestinal regions of mice indicated, dots: individual animals, n: 2–4 mice, mean ± sem. (Right) Representative images of native

*Figure 3 continued on next page*

*Figure 3 continued*

tdTomato fluorescence in intestinal tissue from mouse lines indicated. Scale bars: 100 μm. See ***Figure 3—figure supplements 1–4***.

The online version of this article includes the following source data and figure supplement(s) for figure 3:

**Figure supplement 1.** Determining selectivity of tools for intersectional genetics.

**Figure supplement 2.** Characterization of reporter expression in the oral cavity and airways.

**Figure supplement 3.** Characterization of genetic tools.

**Figure supplement 4.** Differential targeting of enteroendocrine cell types by genetic tools.

**Figure supplement 4—source data 1.** Quantification of differential targeting of enteroendocrine cell types by genetic tools.

(hM3Dq) only in cells expressing both Cre and Flp recombinase (***Sciolino et al., 2016***). Since this approach yielded rare reporter expression in pancreatic islets, we used an additional control mouse line, *Ptf1a-Cre; Vil1-p2a-FlpO (Ptf1a*<sup>INTER</sup>*)*, which targets sparse *Vil1*-expressing pancreatic cells but not intestinal cells (***Figure 4—figure supplement 1***; ***Kawaguchi et al., 2002***).

First, we examined the effect of global enteroendocrine cell activation on gut motility as assessed by movement of charcoal dye following oral gavage. *Neurod1*<sup>INTER</sup>*; inter-hM3Dq-mCherry* mice, *Ptf1a*<sup>INTER</sup>*; inter-hM3Dq-mCherry* mice, and control Cre-negative *Vil1-p2a-FlpO; inter-hM3Dq-mCherry* littermates were injected intraperitoneally (IP) with CNO (fed ad libitum, daytime). After 15 min, charcoal dye was administered, and after an additional 20 min, the gastrointestinal tract was harvested. Charcoal transit distance was calculated by genotype-blinded measurement of the charcoal dye leading edge. In control animals lacking DREADD expression, the leading edge of charcoal dye traversed part of the intestine (littermate controls lacking *Neurod1-Cre*: 22.6 ± 1.2 cm; littermate controls lacking *Ptf1a-Cre*: 22.8 ± 2.0 cm) (***Figure 4***, ***Figure 4—source data 1***). Chemogenetic activation of all enteroendocrine cells in *Neurod1*<sup>INTER</sup>*; inter-hM3Dq-mCherry* mice accelerated gut transit, with the charcoal leading edge traversing 30.8 ± 1.5 cm of the intestine. When DREADD signaling was instead activated in all epithelial cells using *Vil1-Cre; lsl-hM3Dq* mice, gavaged dye failed to enter the intestine at all (***Figure 4—figure supplement 1A***). CNO-accelerated gut transit was not observed *Ptf1a*<sup>INTER</sup>*; inter-hM3Dq-mCherry* mice (22.6 ± 2.6 cm) containing DREADD expression only in pancreatic cells (***Figure 4***, ***Figure 4—figure supplement 1B and C***). Based on these observations, the observed effects in *Neurod1*<sup>INTER</sup>*; inter-hM3Dq-mCherry* mice are due to enteroendocrine cells rather than pancreatic cells, and the net effect of activating all enteroendocrine cells is to promote gut transit.

Next, we examined the effects of activating different enteroendocrine cell subtypes on gut motility. We additionally generated (1) *Pet1*<sup>INTER</sup>*; inter-hM3Dq-mCherry*; (2) *Tac1*<sup>INTER</sup>*; inter-hM3Dq-mCherry*; (3) *Npy1r*<sup>INTER</sup>*; inter-hM3Dq-mCherry*; (4) *Sst*<sup>INTER</sup>*; inter-hM3Dq-mCherry*; (5) *Gip*<sup>INTER</sup>*; inter-hM3Dq-mCherry*; (6) *Cck*<sup>INTER</sup>*; inter-hM3Dq-mCherry*; and (7) *Gcg*<sup>INTER</sup>*; inter-hM3Dq-mCherry* mice, with Cre-negative FlpO-positive *inter-hM3Dq-mCherry* littermates serving as controls (***Figure 4***). As above, CNO was injected (IP) into ad libitum-fed animals followed by oral charcoal gavage. *Pet1*<sup>INTER</sup> cells promoted gut transit (*Pet1*<sup>INTER</sup>: 29.8 ± 1.6 cm, Cre-negative littermates: 22.1 ± 1.5 cm), while *Sst*<sup>INTER</sup> and *Gip*<sup>INTER</sup> cells had no significant effect (*Sst*<sup>INTER</sup>: 24.5 ± 2.0 cm, Cre-negative littermates: 18.8 ± 1.6 cm; *Gip*<sup>INTER</sup>: 23.2 ± 1.0 cm, Cre-negative littermates: 22.0 ± 1.8 cm). Interestingly, single-cell transcriptome data revealed multiple subtypes of enterochromaffin cells, and we observed accelerated gut transit upon chemogenetic activation of *Tac1*<sup>INTER</sup> cells (*Tac1*<sup>INTER</sup>: 36.2 ± 1.4 cm, Cre-negative littermates: 21.0 ± 1.1 cm) but not *Npy1r*<sup>INTER</sup> cells (*Npy1r*<sup>INTER</sup>: 21.8 ± 2.0 cm, Cre-negative littermates: 24.3 ± 2.1 cm). These findings raise the possibility that each enterochromaffin cell subtype may privately communicate with different downstream extrinsic and/or enteric neurons to control gut physiology. In contrast, *Cck*<sup>INTER</sup> and *Gcg*<sup>INTER</sup> cells slowed gut motility (*Cck*<sup>INTER</sup>: 7.1 ± 0.3 cm, Cre-negative littermates: 21.5 ± 2.8 cm; *Gcg*<sup>INTER</sup>: 7.9 ± 0.7 cm, Cre-negative littermates: 22.4 ± 1.9 cm). Ingested food slows gut motility to promote nutrient absorption, while ingested toxins may accelerate gut motility to purge luminal contents (***Nozawa et al., 2009***; ***Van Citters and Lin, 2006***). Consistent with these findings, CCK and GLP1 are released by nutrients while serotonin signaling is required for certain toxin responses (***Drucker, 2016***; ***Gribble and Reimann, 2019***). Simultaneous activation of both pathways, as done in *Neurod1*<sup>INTER</sup>*; inter-hM3Dq-mCherry* mice, masks the slowing of gut transit by *Cck*<sup>INTER</sup> and

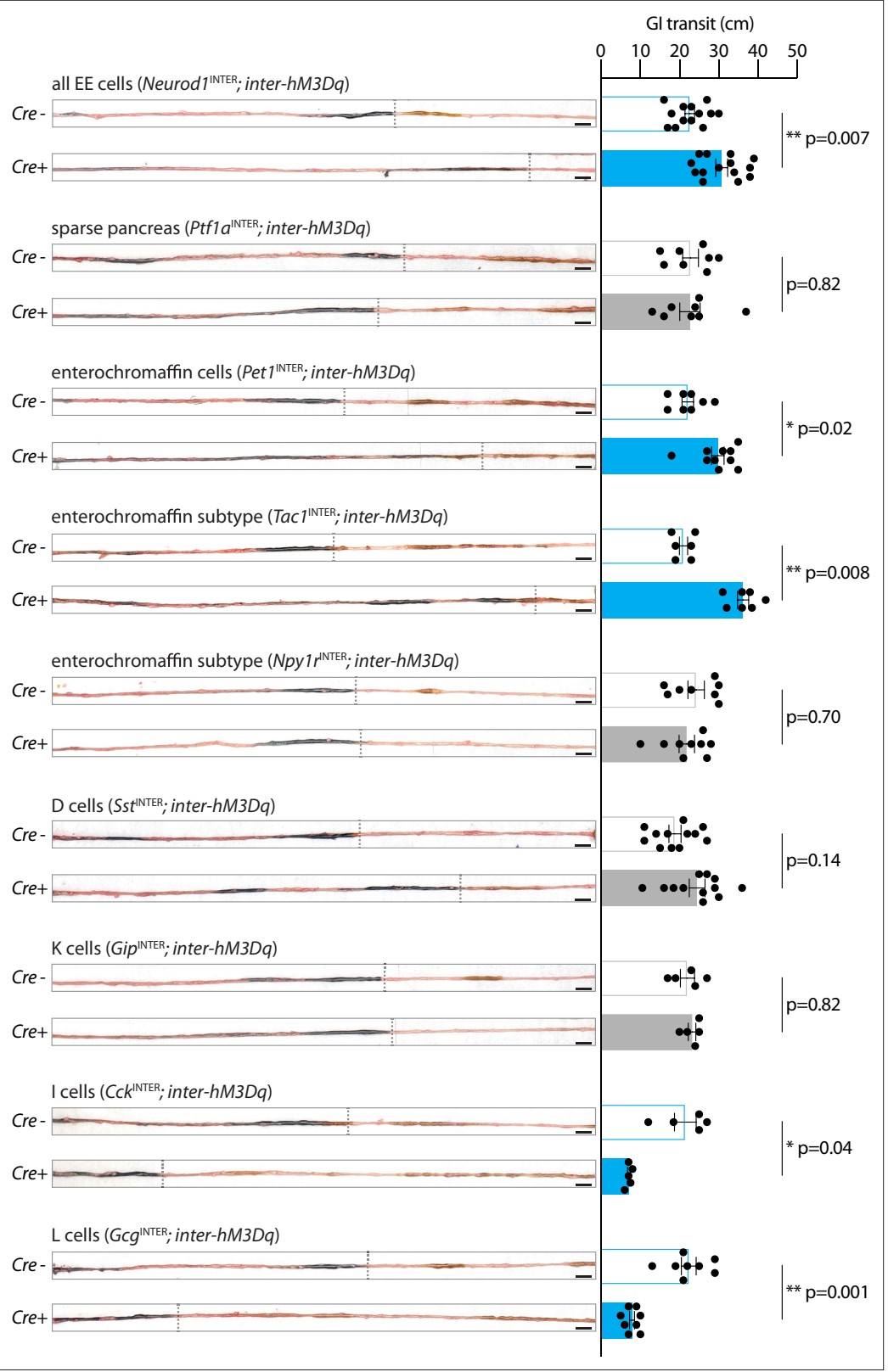

**Figure 4.** Enteroendocrine cell types that accelerate or slow gut transit. Mice of genotypes indicated were injected with CNO (IP, 3 mg/kg) and gavaged orally with charcoal dye. Intestinal tissue was harvested, and the distance between the pyloric sphincter and the charcoal dye leading edge was measured. Representative images (left) and

*Figure 4 continued on next page*

*Figure 4 continued*

quantification (right) of gut transit. Scale bars: 1 cm, circles: individual mice, n: 5–14 mice, mean ± sem, *p<0.05, **p<0.01 by a Mann–Whitney test with Holm–Šídák correction. See *Figure 4—figure supplement 1*.

The online version of this article includes the following source data and figure supplement(s) for figure 4:

**Source data 1.** Quantification of gut transit.

**Figure supplement 1.** Supporting data for gut transit measurements.

---

*Gcg*INTER cells. These findings suggest a hierarchy where neural circuits that mediate toxin responses may achieve priority over those that mediate nutrient responses, at least under conditions of equal and maximal activation. Altogether, we characterize enteroendocrine cell subtypes that have different and sometimes opposing effects on digestive system physiology.

## Enteroendocrine cells that regulate feeding behavior

Next, we examined the effect of global enteroendocrine cell activation on feeding behavior. Fasted mice expressing DREADDs in all enteroendocrine cells (*Neurod1*INTER; *inter-hM3Dq-mCherry*) or in sparse pancreatic cells (*Ptf1a*INTER; *inter-hM3Dq-mCherry*), and their control littermates lacking Cre recombinase, were injected (IP) with CNO and given access to food for 2 hr at dark onset (*Figure 5A*). Animals lacking DREADD expression, or with sparse DREADD expression only in pancreas, ate robustly (~1 g of food over a 2 hr period). In contrast, CNO-induced activation of enteroendocrine cells caused a 26% reduction in food intake (*Figure 5B*, *Figure 5—source data 1*).

To interrogate the roles of different enteroendocrine cell subtypes in feeding regulation, similar experiments were then performed in (1) *Pet1*INTER; *inter-hM3Dq-mCherry*; (2) *Tac1*INTER; *inter-hM3Dq-mCherry*; (3) *Npy1r*INTER; *inter-hM3Dq-mCherry*; (4) *Sst*INTER; *inter-hM3Dq-mCherry*; (5) *Gip*INTER; *inter-hM3Dq-mCherry*; (6) *Cck*INTER; *inter-hM3Dq-mCherry*; and (7) *Gcg*INTER; *inter-hM3Dq-mCherry* mice, with Cre-negative littermates again serving as controls. Chemogenetic activation of enterochromaffin cells reduced feeding behavior (*Figure 5B*, 52.1% reduction). Similar results were seen upon chemogenetic activation of Tac1 and Npy1r cells (*Figure 5—figure supplement 1A*, *Tac1-ires2-Cre*: 48.5% reduction, *Npy1r-Cre*: 79.6% reduction), but we note that these intersectional allele combinations also drove expression in taste cells and rectal epithelium, cell types that could also potentially drive changes in feeding behavior. In contrast, activation of *Sst*INTER and *Gip*INTER cells did not change feeding behavior (*Figure 5B*). Activating *Gcg*INTER cells also reduced feeding (compared to

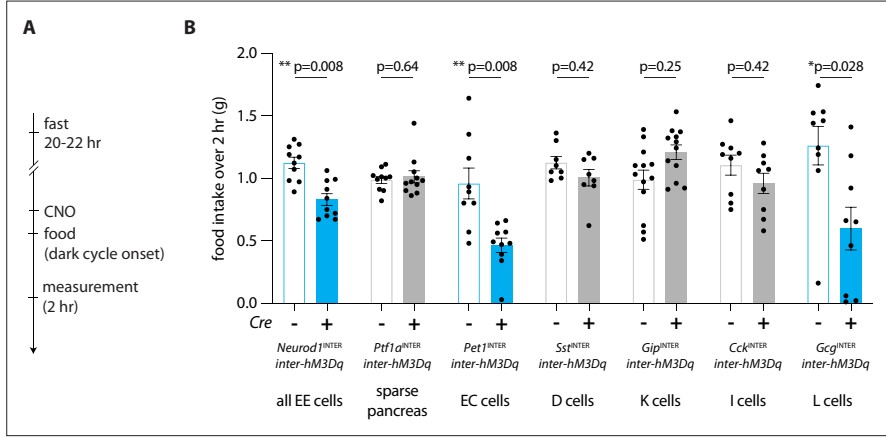

**Figure 5.** Enteroendocrine cell types that reduce feeding. (**A**) Timeline for behavioral assay. (**B**) Mice of genotypes indicated were fasted overnight, injected with CNO (IP, 3 mg/kg), and total food intake was measured during 2 hr ad libitum food access, circles: individual mice, n: 8–13 mice, mean ± sem, *p<0.05 by a Mann–Whitney test with Holm–Šídák correction. See *Figure 5—figure supplement 1*.

The online version of this article includes the following source data and figure supplement(s) for figure 5:

**Source data 1.** Quantification of feeding behavior.

**Figure supplement 1.** Behavioral responses to enteroendocrine cell activation.

**Figure supplement 1—source data 1.** Quantification of feeding behavior.

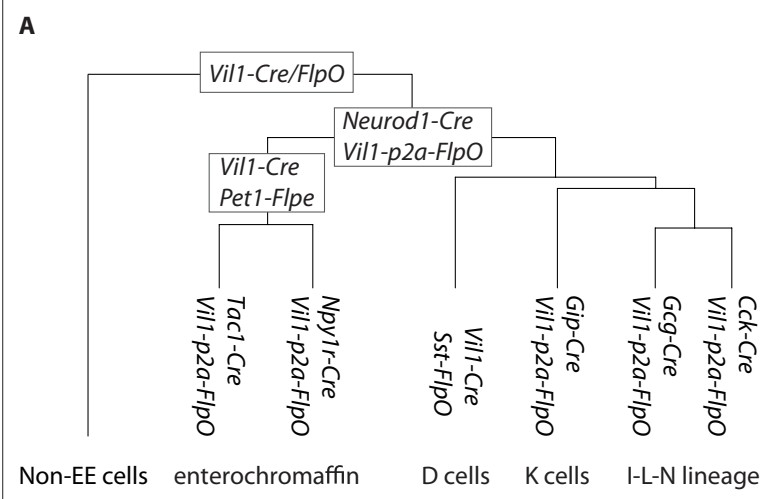

**Figure 6.** Differential regulation of physiology and behavior by enteroendocrine cell subtypes. (**A**) A dendrogram depicting cell types targeted by different genetic tools. (**B**) Summary of feeding and gut transit data obtained for genetic tools that target different enteroendocrine cell types, *only observed in fed state.

Cre-negative littermates, *Gcg*INTER: 52.4% reduction), but surprisingly, activating *Cck*INTER cells lowered feeding only in fed but not fasted mice (***Figure 5—figure supplement 1B***, ***Figure 5—figure supplement 1—source data 1***). This observation is likely due to *Cck-ires-Cre* and *Gcg-Cre* alleles targeting at least partially distinct populations of enteroendocrine cells. Chemogenetic activation of *Gcg*INTER

(single CNO injection) caused a durable reduction of feeding for several hours, with total food intake normalizing by 11 hr, and also evoked a decrease in water intake and the respiratory exchange ratio, but not locomotion (*Figure 5—figure supplement 1C*). For comparison, activating somatostatin cells reduced the respiratory exchange ratio but did not change feeding, water intake, or locomotion. Altogether, we find that some but not all enteroendocrine cells can regulate food intake, and can do so with varying efficacy.

## Conclusion

Here we developed a toolkit involving intersectional genetics for systematic access to each major enteroendocrine cell lineage (*Figure 6A*). We then used chemogenetic approaches to delineate major response pathways of the gut-brain axis (*Figure 6B*). Serotonin-producing enterochromaffin cells express the irritant receptor TRPA1 (*Bellono et al., 2017*) and chemogenetic activation blocks feeding behavior and promotes gut transit, presumably for toxin clearance. Furthermore, different enterochromaffin cell subtypes can have different effects on gut motility, suggesting at least partially nonoverlapping communication pathways with downstream neurons. These findings are consistent with a role for enterochromaffin cells in toxin-induced illness responses, and interestingly, pharmacological blockade of the serotonin receptor HTR3A is a clinical mainstay for nausea treatment (*Freeman et al., 1992*). Other enteroendocrine cell types, including those that produce CCK, GIP, GLP1, neurotensin, and somatostatin, express nutrient receptors yet elicit different physiological and behavioral responses. For example, GLP1 cells slow gut motility, presumably to promote nutrient absorption and decrease feeding behavior (*Gribble and Reimann, 2019*). Additional studies are needed to define gut-brain pathways that mediate nutrient reward, and why receptors for specific nutrients are expressed across a dispersed ensemble of enteroendocrine cells. Together, these experiments provide a highly selective method for accessing enteroendocrine cells in vivo and a direct measure of their various roles in behavior and digestive physiology.

# Materials and methods

**Key resources table**

| Reagent type (species) or resource | Designation | Source or reference | Identifiers | Additional information |
|---|---|---|---|---|
| Strain, strain background (*Mus musculus*) | *Atoh1-Cre* knock-in | *Yang et al., 2010* | | |
| Strain, strain background (*M. musculus*) | *Pet1-FlpE* | *Jensen et al., 2008* | | |
| Strain, strain background (*M. musculus*) | *Ptf1a-Cre* | *Kawaguchi et al., 2002* | | |
| Strain, strain background (*M. musculus*) | *Gip-Cre* | *Svendsen et al., 2016* | | |
| Strain, strain background (*M. musculus*) | *Atoh1-Cre* transgenic | Jax 011104 | | |
| Strain, strain background (*M. musculus*) | *Neurog3-Cre* | Jax 006333 | | |
| Strain, strain background (*M. musculus*) | *Neurod1-Cre* | Jax 028364 | | |
| Strain, strain background (*M. musculus*) | *Sst-ires-Cre* | Jax 013044 | | |
| Strain, strain background (*M. musculus*) | *Sst-ires-FlpO* | Jax 028579 | | |
| Strain, strain background (*M. musculus*) | *Vil1-Cre* | Jax 021504 | | |
| Strain, strain background (*M. musculus*) | *Gcg-Cre* | Jax 030542 | | |

*Continued on next page*

*Continued*

| Reagent type (species) or resource | Designation | Source or reference | Identifiers | Additional information |
|---|---|---|---|---|
| Strain, strain background (*M. musculus*) | Cck-ires-Cre | Jax 012706 | | |
| Strain, strain background (*M. musculus*) | Nts-ires-Cre | Jax 017525 | | |
| Strain, strain background (*M. musculus*) | Mc4r-t2a-Cre | Jax 030759 | | |
| Strain, strain background (*M. musculus*) | Npy1r-Cre | Jax 030544 | | |
| Strain, strain background (*M. musculus*) | Tac1-ires2-Cre | Jax 021877 | | |
| Strain, strain background (*M. musculus*) | Rosa26$^{CAG-lsl-tdTomato}$, Ai14 (lsl-tdTomato) | Jax 007914 | | |
| Strain, strain background (*M. musculus*) | Rosa26$^{CAG-lsl-fsf-tdTomato}$, Ai65 (inter-tdTomato) | Jax 021875 | | |
| Strain, strain background (*M. musculus*) | Rosa26$^{CAG-fsf-eGFP-FLEX-hM3Dq-mCherry}$, (inter-hM3Dq-mCherry) | Jax 026943 | | |
| Strain, strain background (*M. musculus*) | lsl-hM3Dq | Jax 026220 | | |
| Strain, strain background (*M. musculus*) | C57BL/6 | Jax 000664 | | |
| Strain, strain background (*M. musculus*) | Vil1-p2a-FlpO | This paper | | |
| Commercial assay or kit | Chromium single-cell 3' reagent kit v3 | 10X Genomics | | |
| Peptide, recombinant protein | TrypLE express | Thermo Fisher 12604013 | | |
| Other | FBS | VWR 10802-772 | | 5% See 'Single-cell RNA sequencing' |
| Peptide, recombinant protein | DNase | Worthington Biochemical LK003172 | | 100 U/ml |
| Other | TO-PRO-3 | Thermo Fisher T3605 | | 1:10,000 See 'Single-cell RNA sequencing' |
| Other | Calcein Violet | Thermo Fisher 65-0854-39 | | 1:10,000 See 'Single-cell RNA sequencing' |
| Other | Normal donkey serum | Jackson Immuno 017-000-121 | | 5% See 'Tissue histology' section |
| Other | Bovine serum albumin | Jackson Immuno 001-000-161 | | 1% See 'Tissue histology' |
| Other | DAPI Fluoromount-G | Southern Biotech 0100-20 | | See 'Tissue histology' |
| Antibody | Anti-CCK (rabbit polyclonal) | Abcam ab27441 | | 1:1000 |
| Antibody | Anti-CRE (rabbit polyclonal) | Cell Signaling 15036 | | 1:500 |
| Antibody | Anti-GLP1 (rabbit polyclonal) | Novus 2622B MAB10473 | | 1:2000 |
| Antibody | Anti-NTS (rabbit polyclonal) | Immunostar 20072 | | 1:2000 |
| Antibody | Anti-SST (rabbit polyclonal) | Novus 906552 MAB2358 | | 1:1000 |
| Antibody | Anti-5HT (goat polyclonal) | Abcam ab66047 | | 1:2000 |
| Antibody | Donkey anti-rabbit Alexa488 | Jackson Immuno 711-545-152 | | 1:500 |

*Continued on next page*

*Continued*

| Reagent type (species) or resource | Designation | Source or reference | Identifiers | Additional information |
|---|---|---|---|---|
| Antibody | Donkey anti-rabbit AlexaCy3 | Jackson Immuno 711-165-152 | | 1:500 |
| Antibody | Donkey anti-rabbit AlexaCy5 | Jackson Immuno 711-175-152 | | 1:500 |
| Antibody | Donkey anti-rabbit Alexa680 | Thermo Fisher A32802 | | 1:500 |
| Antibody | Donkey anti-goat Alexa488 | Jackson Immuno 705-545-147 | | 1:500 |
| Chemical compound, drug | Clozapine N-oxide dihydrochloride | Fisher Scientific Tocris 6329/10 | | 3 mg/kg |

## Mice

All animal husbandry and procedures were performed in compliance with institutional animal care and use committee guidelines. All animal husbandry and procedures followed the ethical guidelines outlined in the NIH Guide for the Care and Use of Laboratory Animals (https://grants.nih.gov/grants/olaw/guide-for-the-care-and-use-of-laboratory-animals.pdf), and all protocols were approved by the institutional animal care and use committee (IACUC) at Harvard Medical School (protocol #04424). *Atoh1-Cre* knock-in (*Yang et al., 2010*), *Pet1-FlpE* (*Jensen et al., 2008*), *Ptf1a-Cre* (*Kawaguchi et al., 2002*), and *Gip-Cre* (*Svendsen et al., 2016*) mice were described before; *Atoh1-Cre* transgenic (011104), *Neurog3-Cre* (006333), *Neurod1-Cre* (028364), *Sst-ires-Cre* (013044), *Sst-ires-FlpO* (028579), *Vil1-Cre* (021504), *Gcg-Cre* (030542), *Cck-ires-Cre* (012706), *Nts-ires-Cre* (017525), *Mc4rt2a-Cre* (030759), *Npy1r-Cre* (030544), *Tac1-ires2-Cre* (021877), *lsl-tdTomato* (Ai14, Rosa26$^{CAG-lsl-tdTomato}$, 007914), *inter-tdTomato* (Ai65, Rosa26$^{CAG-lsl-fsf-tdTomato}$, 021875), *inter-hM3Dq-mCherry* (Rosa26$^{CAG-fsf-eGFP-FLEX-hM3Dq-mCherry}$, 026943), *lsl-hM3Dq* (026220), and C57BL/6 (000664) mice were purchased (Jackson Laboratory). Both male and female mice between 8 and 24 weeks old were used for all studies, and no differences based on sex were observed. All mice were maintained in the C57BL/6 genetic background. Mouse breeding involved paternal *Cre* alleles, paternal *Flp* alleles, and/or maternal effector genes. *Vil1-Cre* produced occasional germline recombination of *loxP* sites that resulted in ectopic *inter-hM3Dq-mCherry* gene expression; mice with such ectopic expression were excluded based on genotyping of reporter allele DNA extracted from ear tissue with primer 1 (stop cassette forward): atgtctggatctgacatggtaa; primer 2 (*hM3Dq* cassette reverse): tctggagaggagaaattgcca; primer 3 (GFP cassette reverse): ttgaagtcgatgcccttcag; intact allele: ~490 bp, recombined allele: ~290 bp. *Vil1-p2a-FlpO* mice were generated by CRISPR-guided approaches at Boston Children's Hospital Mouse Gene Manipulation Core. Cas9 protein, CRISPR sgRNAs (targeting the stop codon of *Vil1* locus), and an ssDNA (containing a *p2a-FlpO* cassette with 150 bp homology arms) were injected into the pronucleus of C57BL/6 embryos. Founder mice were screened by allele specific PCR analysis with primers flanking the 5′ junction (primer 1: aacagaagttccttaaacaagcca; primer 2: aacaggaactggtacagggtcttg; ~930 bp), FlpO internally (primer 1: acaagggcaacagccaca; primer 2: tcagatccgcctgttgatgt; ~830 bp), and the 3′ junction (primer 1: accccctggtgtacctgga; primer 2: tagccctccctttttgagtgtga; ~840 bp), followed by Sanger sequencing to validate the allele. Selected *Vil1-p2a-FlpO* founder mice were viable, fertile, and back crossed to C57BL/6 mice for at least three generations.

## Single-cell RNA sequencing

Enteroendocrine cells were acutely harvested using a protocol modified from previous publications (*Haber et al., 2017*; *Sato et al., 2009*). Intestinal tissue was obtained from *Neurog3-Cre; lsl-tdTomato* mice (one adult male), or *Neurod1-Cre; lsl-tdTomato* (three adult females), cut longitudinally, washed (cold phosphate-buffered saline [PBS]), cut into small ~5 mm pieces, and incubated (gentle agitation, 20 min, 4°C) in EDTA solution (20 mM EDTA-PBS, Ca/Mg-free) in LoBind Protein tubes (Eppendorf 0030122216). The specimen was shaken, the tissue allowed to settle, and the supernatants collected. The residual tissue was again incubated similarly with EDTA solution, and supernatants were combined, and centrifuged (300 × *g*, 5 min, 4°C) Pellets were washed (2×, PBS [Ca/Mg-free] supplemented with 5% fetal bovine serum [FBS], 4°C) and incubated (37°C, 2 min) in protease solution (TrypLE express,

Thermo Fisher 12604013) supplemented with DNase (100 U/ml, Worthington Biochemical LK003172). The suspension containing dissociated cells was centrifuged (300 × $g$, 5 min), washed (2×, PBS [Ca/Mg-free] containing 5% FBS, 4°C) The resulting pellet was resuspended in FACS buffer (5% FBS in DMEM/F12, HEPES, no phenol red) containing DNase (100 U/ml), TO-PRO-3 (Thermo Fisher T3605, 1:10,000) to label dead cells, and Calcein Violet (Thermo Fisher 65-0854-39, 1:10,000) to label living cells. Cells were filtered (1 × 70 um, 1 × 40 um) and tdTomato+, Calcein Violet+, TO-PRO-3- cells were collected by fluorescence activated cell sorting using a FACS Aria (BD Biosciences). Collected cells were then loaded into the 10X Genomics Chromium Controller, and cDNA prepared and amplified according to manufacturer's protocol (10X Genomics, Chromium single-cell 3′ reagent kit v3, 12 cycles per amplification step). The resulting cDNA was sequenced on a NextSeq 500 at the Harvard Medical School Biopolymers Facility. Sequence reads were aligned to the mm39 mouse transcriptome reference, and feature barcode matrices were generated using 10X Genomics CellRanger. Unique transcript (UMI) count matrices were analyzed in R v4.1.1 using Seurat v4.0.5 (*Beutler et al., 2017*; *Satija et al., 2015*). The cell barcodes were filtered, removing cells with a high number of UMIs (>125,000) or high percentage of mitochondrial genes (>25%). The filtered UMI count matrix was transformed using SCTransform (*Hafemeister and Satija, 2019*). Transformed matrices from *Neurog3* and *Neurod1* samples were integrated (nFeature = 3000), and integrated matrices used for cluster identification and UMAP projections. Additional clusters of low-quality cells (defined by low-average UMI counts and low-average feature counts across the cluster) were removed. To examine the diversity among enteroendocrine cells, cell barcodes belonging to enteroendocrine cells from *Neurog3* and *Neurod1* samples were identified and reanalyzed separately. Matrices of enteroendocrine cells from *Neurog3* and *Neurod1* samples were transformed and integrated (nFeature = 3000). Differential gene expression (Wilcoxon ranked-sum test) was conducted on UMI counts matrices that were log normalized and scaled. Seurat's BuildClusterTree function was used to spatially arrange clusters based on relative similarity in gene expression. Two serotonergic clusters were merged *post hoc* (to become cluster EC_3) due to the absence of any single signature gene that effectively distinguished them. Gene expression data in all UMAP plots is shown as a natural log of normalized UMI counts. Further details and full parameters of analysis will be provided on GitHub upon publication: https://github.com/jakaye/EEC_scRNA, copy archived at (*Hayashi, 2023*).

## Tissue histology

For histology, mice were perfused intracardially with PBS and then fixative (4% paraformaldehyde/PBS). Intestinal regions and other organs were dissected (duodenum: first 2 cm after the pyloric sphincter, jejunum: middle 2 cm, ileum: last 2 cm before the cecum, colon: first 2 cm after the cecum, and rectum: last 2 cm accessible via the pelvic cavity) and postfixed (1–2 hr, 4°C). Samples were then incubated in 30% sucrose/PBS (overnight, 4°C), embedded in Tissue-Tek OCT, frozen, cryosectioned, and placed on glass slides. Slides were incubated with primary antibodies at dilutions indicated below (overnight, 4°C, PBS supplemented with 0.05% Tween20, 0.1% TritonX, and either 5% normal donkey serum or 1% BSA) and then with fluorophore-conjugated secondary antibodies (1:500, 2 hr, RT). Sections were mounted (DAPI Fluoromount-G, Southern Biotech 0100-20), coverslipped, and imaged using a Nikon A1R confocal microscope, an Olympus FV1000 confocal microscope, or a Zeiss Axiozoom V16 fluorescent stereoscope. Microscope images are presented as z-projections. Quantification of tdTomato expression and antibody staining was performed manually using a Nikon Ti2 inverted microscope. Antibodies were rabbit anti-CCK (Abcam ab27441, 1:1000), rabbit anti-CRE (Cell Signaling 15036, 1:500), rabbit anti-GLP1 (Novus 2622B MAB10473, 1:2000), rabbit anti-NTS (Immunostar 20072, 1:2000), rabbit anti-SST (Novus 906552 MAB2358, 1:1000), goat anti-5HT (Abcam ab66047, 1:2000), donkey anti-rabbit Alexa488, Cy3, Cy5, Alexa680 (Jackson Immuno Research, Thermo Fisher, 1:500), donkey anti-goat Alexa488 (Jackson Immuno Research, 1:500).

## Gut transit measurements

DREADD-expressing and control animals (ad libitum fed) were injected with CNO (3 mg/kg, IP). After 15 min, charcoal dye (200 µl, 10% activated charcoal, 10% gum Arabic in water), or for *Figure 4—figure supplement 1*, carmen red dye (200 µl, 6% carmen red, 0.5% methyl cellulose in water), was gavaged orally, and 20 min later, mice were euthanized and the gastrointestinal tract was harvested. The distance between the pyloric sphincter and the charcoal dye leading edge was measured by an

observer blind to animal genotype. All animals were naive to CNO exposure, except for some *Gip-Cre* mice due to limited availability of mice.

## Feeding measurements

Experimental mice were individually housed for 3 days and habituated to feeding from a ceramic bowl. Animals were either fed ad libitum or fasted for the last 20–22 hr in a new clean cage with some bedding material from the previous cage. CNO was injected (3 mg/kg, IP), and food pellets presented 15 min later at the onset of darkness. Food intake was measured over the course of 2 hr by weighing the amount of residual food, with genotypes revealed *post hoc* to achieve a genotype-blinded analysis. Studies involved fasted mice that were naive to prior CNO exposure or fed mice that were either naive to CNO or acclimated for at least a week after prior CNO exposure.

## Body composition and indirect calorimetry

Body composition (lean mass and fat mass) was first analyzed for each experimental group with a 3-in-1 Echo MRI Composition Analyzer (Echo Medical Systems, Houston, TX), and no significant differences were observed. Animals were then placed in a Sable Systems Promethion indirect calorimeter maintained at 23°C ± 0.2°C. Mice were singly housed in metabolic cages with corn cob bedding and ad libitum access to Labdiet 5008 chow (56.8/16.5/26.6 carbohydrate/fat/protein). After 18 hr, all mice were injected with PBS (IP) for acclimatation to handling and mild injection stress. The following day, mice were injected with CNO (3 mg/kg, IP) approximately 30 min before dark onset. Animals were then analyzed for food and water consumption, body weight, distance traveled, and respiratory exchange ratio. Statistical analysis was performed with CalR (*Mina et al., 2018*).

## Statistical analysis

Graphs represent data as mean ± sem, as indicated in figure legends. All data points were derived from different mice except some mice in *Figure 4* (*Gip-Cre; Vil1-p2a-FlpO; inter-hM3Dq-mCherry* mice) were previously used in feeding assays and some mice in *Figure 5—figure supplement 1* (*Ptf1a-Cre; Vil1-p2a-FlpO; inter-hM3Dq-mCherry*: 21/21 mice, *Cck-ires-Cre; Vil1-p2a-FlpO; inter-hM3Dq-mCherry*: 7/19 mice, *Gip-Cre; Vil1-p2a-FlpO; inter-hM3Dq-mCherry*: 10/10 mice, and *Vil1-Cre; Sst-ires-FlpO; inter-hM3Dq-mCherry*: 9/16 mice) were previously used in prior feeding assays for *Figure 5*. When mice were reused, they were acclimated for at least a week after prior CNO exposure.

Sample sizes (from left to right): *Figure 3* (Pet1: 4, 4, 3, 3, 3; Sst: 4, 4, 3, 2, 4; Gip: 4, 4, 4, 2, 4; Cck: 4, 4, 3, 4, 4; Gcg: 3, 3, 3, 2, 3), *Figure 3—figure supplement 4A* (serotonin antibody: 17, 17, 16, 10, 13; Sst antibody: 18, 18, 12, 10, 13; Cck antibody: 18, 18, 12, 13, 13; GLP1 antibody: 18, 18, 13, 10, 13), B (serotonin antibody: 3, 3, 3, 3, 3; Sst antibody: 4, 4, 3, 3, 3; Cck antibody: 3, 3, 3, 3, 3; GLP1 antibody: 3, 3, 3, 3, 3), C (serotonin antibody: 4, 4, 4, 2, 4; Sst antibody: 4, 4, 4, 2, 4; Cck antibody: 4, 4, 4, 2, 4; GLP1 antibody: 4, 4, 4, 2, 4), D (serotonin antibody: 4, 4, 4, 2, 4; Sst antibody: 4, 4, 4, 2, 4; Cck antibody: 4, 4, 4, 2, 4; GLP1 antibody: 4, 4, 4, 2, 4), E (serotonin antibody: 3, 3, 2, 3, 3; Sst antibody: 3, 3, 2, 3, 3; Cck antibody: 4, 4, 2, 3, 3; GLP1 antibody: 4, 4, 2, 3, 3), F (serotonin antibody: 3, 3, 2, 3, 3; Sst antibody: 3, 3, 2, 3, 3; Cck antibody: 3, 3, 2, 3, 3; GLP1 antibody: 3, 3, 2, 3, 3), *Figure 4* (13, 14, 8, 8, 8, 10, 6, 7, 8, 9, 12, 12, 5, 5, 5, 8, 8, 8), *Figure 4—figure supplement 1A* (4, 5), *Figure 5* (10, 10, 10, 11, 9, 10, 8, 8, 13, 12, 9, 9, 9, 9), *Figure 5—figure supplement 1A* (4, 7, 6, 6,), B (9, 12, 9, 8, 4, 6, 9, 10), *Figure 5—figure supplement 1C* (7, 9, 3, 5).

Statistical significance was measured using a Mann–Whitney test with Holm–Šídák correction on Prism 9 (GraphPad) for *Figure 4*, *Figure 5*, and *Figure 5—figure supplement 1A and B*, a Mann–Whitney test on Prism 9 (GraphPad) for *Figure 4—figure supplement 1A*, and ANCOVA and ANOVA on CalR for *Figure 5—figure supplement 1C* (*Mina et al., 2018*).

## Source data

The source data Excel file contains raw numerical data used for all bar graphs and statistical analyses. Single-cell transcriptome data is available with a GEO GSE accession number GSE224223.

## Materials availability statement

*Vil1-p2a-FlpO* mice will be deposited in Jackson Laboratory and made generally available upon reasonable request.

## Declaration of interest

SDL and FMG are consultants for Kallyope, Inc.

## Acknowledgements

We thank Nancy Thornberry and Paul Richards for manuscript comments, Lin Gan (Rochester) for *Atoh1-Cre* mice, Chris Wright (Vanderbilt) for *Ptf1a-Cre* mice, Patricia Jensen (NIH) and Nicholas Plummer (NIH) for genotyping advice, members of the Liberles laboratory for experimental advice and assistance, the Boston Children's Hospital Mouse Gene Manipulation Core (NIH P50 HD105351) for help with generating mice, the Beth Israel Deaconess Medical Center Energy Balance Core (OD028635, P30 DK034854) for help with CLAMS assays, the Harvard Medical School Nikon Imaging Center and Neurobiology Imaging Facility (NINDS P30 NS072030) for help with microscopy, the Biopolymers Facility for help with sequencing, the Immunology Flow Cytometry Core Facility for help with cell sorting, and the Harvard Medical School O2 High Performance Computer Cluster for bioinformatics support. This work was supported by the NIH (DP1AT009497 to SDL, R01DK103703 to SDL, and T32 HL007901 to JAK), and the Food Allergy Science Initiative (to SDL). MH is a fellow of Japan Society for the Promotion for Science. SDL is an investigator of the Howard Hughes Medical Institute.

## Additional information

### Competing interests

Fiona M Gribble: Is a consultant for Kallyope, Inc. Stephen D Liberles: Reviewing editor, eLife. The other authors declare that no competing interests exist.

### Funding

| Funder | Grant reference number | Author |
| --- | --- | --- |
| Food Allergy Initiative | | Stephen D Liberles |
| National Institutes of Health | DP1AT009497 | Stephen D Liberles |
| National Institutes of Health | R01DK103703 | Stephen D Liberles |
| Howard Hughes Medical Institute | P30 NS072030 | Stephen D Liberles |
| Japan Society for the Promotion of Science | | Marito Hayashi |
| National Institutes of Health | T32 HL007901 | Judith A Kaye |
| Wellcome grant | 220271/Z/20/Z | Fiona M Gribble |
| Medical Research Council | MRC_MC_UU_12012/3 | Fiona M Gribble |

The funders had no role in study design, data collection and interpretation, or the decision to submit the work for publication.

### Author contributions

Marito Hayashi, Conceptualization, Resources, Data curation, Formal analysis, Investigation, Visualization, Methodology, Writing – original draft, Writing – review and editing; Judith A Kaye, Data curation, Software, Formal analysis; Ella R Douglas, Narendra R Joshi, Investigation; Fiona M Gribble, Frank Reimann, Resources, Writing – review and editing; Stephen D Liberles, Conceptualization, Supervision, Funding acquisition, Writing – original draft, Writing – review and editing

### Author ORCIDs

Marito Hayashi  http://orcid.org/0000-0002-9367-6389
Narendra R Joshi  http://orcid.org/0000-0002-5030-389X
Fiona M Gribble  http://orcid.org/0000-0002-4232-2898

Frank Reimann http://orcid.org/0000-0001-9399-6377
Stephen D Liberles http://orcid.org/0000-0002-2177-9741

### Ethics

All animal husbandry and procedures were performed in compliance with institutional animal care and use committee guidelines. All animal husbandry and procedures followed the ethical guidelines outlined in the NIH Guide for the Care and Use of Laboratory Animals (https://grants.nih.gov/grants/olaw/guide-for-the-care-and-use-of-laboratory-animals.pdf), and all protocols were approved by the institutional animal care and use committee (IACUC) at Harvard Medical School.

### Decision letter and Author response

Decision letter https://doi.org/10.7554/eLife.78512.sa1
Author response https://doi.org/10.7554/eLife.78512.sa2

## Additional files

### Supplementary files
• MDAR checklist

### Data availability

The source data excel file contains raw numerical data used for all bar graphs and statistical analyses. Single-cell transcriptome data are available at NCBI Gene Expression Omnibus with accession GSE224223.

The following dataset was generated:

| Author(s) | Year | Dataset title | Dataset URL | Database and Identifier |
|---|---|---|---|---|
| Hayashi M, Kaye JA, Douglas ER, Joshi NR, Gribble F, Reimann F, Liberles SD | 2023 | Enteroendocrine cell lineages that differentially control feeding and gut motility | https://www.ncbi.nlm.nih.gov/geo/query/acc.cgi?acc=GSE224223 | NCBI Gene Expression Omnibus, GSE224223 |

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
