## [Editor Report]

As digested food moves through the intestines specialized epithelial cells (called Enterochromaffin Cells or EECs) sense and respond to the constituent chemicals. The current work utilizes single-cell transcriptomic analyses and intersectional approaches to define and genetically manipulate subsets of EECs. Key findings are that direct stimulation of EEC subtypes influences key aspects of feeding, specifically gut transit, ingestion, and food preference.

---

## [Decision Letter]

**Decision letter after peer review:**

Thank you for submitting your article "Enteroendocrine cell lineages that differentially control feeding and gut motility" for consideration by *eLife*. Your article has been reviewed by 3 peer reviewers, including Alexander Theodore Chesler as Reviewing Editor and Reviewer #1, and the evaluation has been overseen by K VijayRaghavan as the Senior Editor.

Essential revisions:

In general, the work provides new tools and insight into the function of enterochromaffin cells (EECs). However, several key points were raised by reviewers that we would like to see addressed in a revised manuscript:

(1) Better characterization of the precise cells in which recombination takes place and more information about the rostral-caudal expression patterns.

(2) Controls for the lack of specificity of DREADD mediated cell manipulation, particularly for the Npy1r-line.

(3) Improved controls for both feeding behavior and particularly for the conditioned preference assay. These could include receptor antagonists to validate their hM3Dq activation approach and better link EEC subtypes to differential functions.

The complete reviews are provided below. We hope these are helpful in preparing your revision. When resubmitting, please include point-by-point responses and highlight the changes to the manuscript.

*Reviewer #1 (Recommendations for the authors):*

The manuscript could be rewritten for clarity. For example, I found it hard to keep track of the crosses since the full genetic names are used throughout.

Specific comments (by figure and associated supplements)

Figure 1:

a. The images of the genetic marker labeling could be clearer. Nicer overview sections and more detailed zoom-in panels would be helpful. EECs are small, so they are hard to see in the images provided.

b. What are the cells in the pancreas?

c. Why is the tongue, epiglottis, and trachea in the supplement when these could be consequential? Most of the figure is blank sections, not sure about the need to take up so much space, maybe move these to supplemental?

d. Figure 1B is slightly misleading as a complete survey of the body is not presented and so labeling outside the gut could be occurring in the Vil1 and NeuroD1/Vil1 strains.

Figure 2:

It would have been helpful to have the panel of sequencing data in 2D be with the markers used in the crosses for the rest of the study and associated directly with 2E.

Figure 3:

a. Please provide zoom-in panels for 3A so we can see the EEC cells and the overlap with the associated markers.

b. Figure 3, supplement 2 is critical data, especially for Npy1r-Cre/Vil-FlpO. More details as to what is being labeled in these areas would be very helpful for interpreting behavioral data from this strain.

c. Figure 3, supplement 3B is important data for interpreting phenotypes. I assume chemogenetic activation of colon cells could have effects on behavior? Can these data be moved to the main figure and can the function of these cells be tested?

Figure 4:

This is a nice assay, and the presentation is stunning and convincing.

Figure 5:

It is strange that many of the EEC subtypes have no effect. One wonders if relying solely on this assay has revealed the functions of these cells. Of particular concern is that the Npy1r cross has the biggest effect, and this is the strain with expression in the tongue, epiglottis, and trachea.

Figure 6:

Generally, these data are the least convincing and have a lot of variation. Not clear what conclusions can be drawn from this. The LiCl should be in the main figure. A positive reinforcing stimulus might also be included.

*Reviewer #2 (Recommendations for the authors):*

Overall, this is a well-written report. The Villin-Flp mouse is a great approach for manipulating EEC subtypes and it was convincingly shown to be highly selective to the gut. It was put to good use in characterizing gut motility and feeding behaviour.

*Reviewer #3 (Recommendations for the authors):*

1. Figure 3C contains important but rather incomplete data (perhaps reflecting antibody availability) about a few driver lines; as few as 14 cells and sections from 2 animals were used for generating these data. Some of this information is also repeated in the text (along with data for other lines not shown here). It is crucial for accurate interpretation that data are extended to better characterize the lines and that the data are supplied as a figure that the reader can understand. These data will provide a critical framework for interpreting all the subsequent functional data.

2. It appears that only very few EECs are labeled using the Npy1r-Cre driver line (5% of serotonergic cells) despite sc-data that suggests quite a broad expression. Particularly since this line is also active outside the gut when intersected with villin-Flp, behavioral data using the chemogenetic strategy appear poorly controlled, and conclusions about feeding suppression are risky.

3. Descriptions of expression given in Figure 4 need more validation; e.g., Tac1 and Fev are expressed in the sc-population called precursor cells and are therefore likely to both target the same broad group of ECCs.

4. Feeding behavior data Figure 5 suggests great inter-animal variation for some groups and consistency for others meaning that the analysis is likely underpowered. For example, the 4 Tac1 control mice all eat at the high end of the range. The Npy1r data are unexpected given the poor targeting of ECCs by this line, but expression in taste receptor cells cannot be discounted for this CNO-driven feeding deficit. Finally, all data presented about Cck and Gcg-Cre lines suggests broad overlap in targeted EECs making the discrepancies between lines difficult to explain.

5. The assay data presented in Figure 6 are just too scattered for any conclusions to be drawn. Indeed, it would appear that the only "significant" difference with CNO treatment arises not from the effect of CNO but by virtue of a difference in the control mice.

6. Some discussion of how the single-cell data presented here compare with previously published results would be appropriate. In addition, the main figure should include expression patterns for genes used as driver lines (violin plots); it would also be helpful to have feature plots for these genes in a supplement. It is important that some of the data e.g., expression of Slc5a1 and Ffars as well as lack of expression of Piezo2 are tested by e.g., ISH if they deserve a section of text; otherwise this section could be deleted since it is not fundamental to the story.

7. The early focus on lines that are non-specific or not a feature of this paper (including much of Figure 1) and the demonstration of lots of negative expression outside the gut (as well as expression of the Cre-drivers alone) are not really relevant to the story and should be dramatically shortened providing room for more fundamental controls.

8. Many strategies are relatively commonplace but are described in great detail, similarly, the nomenclature for the lines needs to be simplified to make this paper more straightforward to read.

---

## [Author Response]

Essential revisions:(1) Better characterization of the precise cells in which recombination takes place and more information about the rostral-caudal expression patterns.

For each intersectional allele combination, we performed extensive two-color analysis to compare expression patterns of reporters and native hormones. Furthermore, these studies were performed at five discrete positions along the proximal-distal axis of the intestine. This data set highlights key spatial differences in hormone expression; for example, one cell type marked by GIP (in *Gip-Cre; Vil1-p2a-FlpO* mice) is predominantly expressed in proximal intestine while another cell type marked by GLP-1 (in *Gcg-Cre; Vil1-p2a-FlpO* mice) is instead enriched more distally. These new findings are presented in revised Figure 3 and Figure 3-Supplement 4.

(2) Controls for the lack of specificity of DREADD mediated cell manipulation, particularly for the Npy1r-line.

We agree with reviewers' concerns about the *Npy1r-Cre* line. *Npy1r-Cre* and *Vil1-p2a-FlpO* both target taste cells and broadly label the colon epithelium. Since expression outside of enteroendocrine cells persists, we have de-emphasized it in the text. We moved corresponding behavioral data to Supplementary Figures, toned back interpretations, and openly discussed caveats related to expression in other cell types, including in the summary table of Figure 6.

(3) Improved controls for both feeding behavior and particularly for the conditioned preference assay. These could include receptor antagonists to validate their hM3Dq activation approach and better link EEC subtypes to differential functions.

We decided to remove all data in the text related to conditioned flavor avoidance. We found that in our paradigm, activation of enterochromaffin cells causes such a powerful avoidance-teaching signal that aversion generalizes to other novel odors. This results in decreased total drinking during the test day (data from a lickometer is presented in Author response image 1), which impairs measurement of flavor discrimination, and is not a standard way to present data from this assay. We spent a great deal of time attempting to optimize the flavor avoidance assay for enterochromaffin cells, and this remains ongoing; we note that large cohorts of mice containing three knock-in alleles are needed, which is part of why it has taken so long. Out of an abundance of caution, we decided to remove these data entirely and explore further in a future study.

**Author response image 1. sa2fig1:** 

Reviewer #1 (Recommendations for the authors):The manuscript could be rewritten for clarity. For example, I found it hard to keep track of the crosses since the full genetic names are used throughout.

We have simplified the terminology by replacing "*gene-Cre; Vil1-p2a-FlpO*" with "*gene*^INTER^", which we think will be more intuitive.

Specific comments (by figure and associated supplements)Figure 1:a. The images of the genetic marker labeling could be clearer. Nicer overview sections and more detailed zoom-in panels would be helpful. EECs are small, so they are hard to see in the images provided.

We included zoom-in panels in Figure 1 as suggested.

b. What are the cells in the pancreas?

*Vil1-p2a-FlpO* labels rather sparse cells both within and outside of islets, and does not cleanly or uniformly hit a single pancreatic cell type. It perhaps reflects a very low frequency of recombination in an early pancreatic progenitor. Nevertheless, chemogenetic activation of pancreatic cells labeled in *Ptf1a-Cre; Vil1-p2a-FlpO* mice (*Ptf1a* is a developmental marker that broadly labels pancreatic cells) had no effects on gut motility or feeding (Figures 4 and 5), indicating that this off-target labeling does not explain effects observed from enteroendocrince cell activation.

c. Why is the tongue, epiglottis, and trachea in the supplement when these could be consequential? Most of the figure is blank sections, not sure about the need to take up so much space, maybe move these to supplemental?

We moved some of the data around as suggested, including moving data for the tongue to the main figure (Figure 1D), and some negative images to Figure 1—figure supplement 1. We do think some of the negative images from the intersectional lines are important to illustrate how selective the genetic tools are, and we left some key ones in the main figure.

d. Figure 1B is slightly misleading as a complete survey of the body is not presented and so labeling outside the gut could be occurring in the Vil1 and NeuroD1/Vil1 strains.

This is just a conceptual cartoon to illustrate our strategy. We are pretty transparent about where we see signal in each line.

Figure 2:It would have been helpful to have the panel of sequencing data in 2D be with the markers used in the crosses for the rest of the study and associated directly with 2E.

Thank you- we now include single-cell transcriptional data for all Cre line markers used later in the study; these genes are highlighted with dashed boxes in the violin plots of Figure 2.

Figure 3:a. Please provide zoom-in panels for 3A so we can see the EEC cells and the overlap with the associated markers.b. Figure 3, supplement 2 is critical data, especially for Npy1r-Cre/Vil-FlpO. More details as to what is being labeled in these areas would be very helpful for interpreting behavioral data from this strain.c. Figure 3, supplement 3B is important data for interpreting phenotypes. I assume chemogenetic activation of colon cells could have effects on behavior? Can these data be moved to the main figure and can the function of these cells be tested?

Figure 3 has been completely re-done, and includes zoom-in panels as requested. As detailed in the intro, we provide more information about reporter expression in taste cells and colon of *Npy1r-Cre; Vil1-p2a-FlpO* (and for colon, *Tac1-Cre; Vil1-p2a-FlpO*) mice. We share the reviewer's concern about this particular intersectional pair, and have de-emphasized it in the text. We moved corresponding behavioral data to Supplementary Figures, toned back interpretations, and openly discussed caveats related to expression in other cell types.

Figure 4:This is a nice assay, and the presentation is stunning and convincing.

Thank you!

Figure 5:It is strange that many of the EEC subtypes have no effect. One wonders if relying solely on this assay has revealed the functions of these cells. Of particular concern is that the Npy1r cross has the biggest effect, and this is the strain with expression in the tongue, epiglottis, and trachea.

We agree that there are almost certainly roles for EECs in other contexts, like metabolism. For example, in the updated Figure 5 figure supplement 1C, we show that the acute activation of Sst subtype reduces RER, impacting metabolism. We aim to expand our repertoire of behavioral and physiological readouts for EEC activation in future studies.

Figure 6:Generally, these data are the least convincing and have a lot of variation. Not clear what conclusions can be drawn from this. The LiCl should be in the main figure. A positive reinforcing stimulus might also be included.

As detailed above, we decided to remove all data in the text related to conditioned flavor avoidance. Activation of enterochromaffin cells causes such a powerful avoidance-teaching signal that aversion generalizes to other novel odors. This results in decreased total drinking during the test day (data from a lickometer is presented in Author response image 1), which impairs measurement of flavor discrimination, and is not a standard way to present data from this assay. We spent a great deal of time attempting to optimize the flavor avoidance assay for enterochromaffin cells, and this remains ongoing; we note that large cohorts of mice containing three knock-in alleles are needed, which is part of why it has taken so long. Out of an abundance of caution, we decided to remove these data entirely and explore further in a future study.

Reviewer #3 (Recommendations for the authors):1. Figure 3C contains important but rather incomplete data (perhaps reflecting antibody availability) about a few driver lines; as few as 14 cells and sections from 2 animals were used for generating these data. Some of this information is also repeated in the text (along with data for other lines not shown here). It is crucial for accurate interpretation that data are extended to better characterize the lines and that the data are supplied as a figure that the reader can understand. These data will provide a critical framework for interpreting all the subsequent functional data.

We have extensively revamped Figure 3, performing an extensive two-color analysis to characterize each Cre/Flp intersectional tool across the entire proximal-to-distal axis of the intestine. We think this analysis is far more rigorous than any existing characterization in the field of genetic tools for enteroendocrine cells. Furthermore, genetic control reported here is far more selective than other studies (some in major journals) that do not use intersectional lines and have far broader expression patterns.

2. It appears that only very few EECs are labeled using the Npy1r-Cre driver line (5% of serotonergic cells) despite sc-data that suggests quite a broad expression. Particularly since this line is also active outside the gut when intersected with villin-Flp, behavioral data using the chemogenetic strategy appear poorly controlled, and conclusions about feeding suppression are risky.

We agree with reviewers' concerns about the *Npy1r-Cre* and *Tac1-Cre* line. *Npy1r-Cre* and *Vil1-p2a-FlpO* both target taste cells and broadly label the colon epithelium. We note that the 5% overlap is from duodenum, but our single-cell seq analysis was performed on cells obtained throughout the small intestine, and the percentage overlap increases substantially in distal intestine towards the colon. Since expression outside of enteroendocrine cells persists, we have de-emphasized it in the text. We moved corresponding behavioral data to Supplementary Figures, toned back interpretations, and openly discussed caveats related to expression in other cell types, including in the summary table of Figure 6.

3. Descriptions of expression given in Figure 4 need more validation; e.g., Tac1 and Fev are expressed in the sc-population called precursor cells and are therefore likely to both target the same broad group of ECCs.

Two-color analysis indicated that Tac1-Cre marks ~80% of serotonin-expressing cells (page 11, line 22), so there is certainly a high degree of overlap as the reviewer suggests. As detailed above, we have de-emphasized feeding data using the two Cre lines that target subtypes of serotonin cells (Tac1 and Npy1r) since expression in the colon persists in both of these lines.

4. Feeding behavior data Figure 5 suggests great inter-animal variation for some groups and consistency for others meaning that the analysis is likely underpowered. For example, the 4 Tac1 control mice all eat at the high end of the range. The Npy1r data are unexpected given the poor targeting of ECCs by this line, but expression in taste receptor cells cannot be discounted for this CNO-driven feeding deficit. Finally, all data presented about Cck and Gcg-Cre lines suggests broad overlap in targeted EECs making the discrepancies between lines difficult to explain.

As discussed for the prior point, we de-emphasized feeding data using the two Cre lines that target subtypes of serotonin cells (Tac1 and Npy1r). Cck and Gcg share a developmental lineage, so the partial- but not complete- overlap reflects the inherent biology of the system. Differences in the feeding assay almost certainly reflect differential cell targeting by these Cre lines. We agree the data is surprising, but is nevertheless what we observed across three independently repeated experiments for each cell subtype.

5. The assay data presented in Figure 6 are just too scattered for any conclusions to be drawn. Indeed, it would appear that the only "significant" difference with CNO treatment arises not from the effect of CNO but by virtue of a difference in the control mice.

As detailed above, we decided to remove all data in the text related to conditioned flavor avoidance. Activation of enterochromaffin cells causes such a powerful avoidance-teaching signal that aversion generalizes to other novel odors. This results in decreased total drinking during the test day (data from a lickometer is presented in Author response image 1), which impairs measurement of flavor discrimination, and is not a standard way to present data from this assay. We spent a great deal of time attempting to optimize the flavor avoidance assay for enterochromaffin cells, and this remains ongoing; we note that large cohorts of mice containing three knock-in alleles are needed, which is part of why it has taken so long. Out of an abundance of caution, we decided to remove these data entirely and explore further in a future study.

6. Some discussion of how the single-cell data presented here compare with previously published results would be appropriate. In addition, the main figure should include expression patterns for genes used as driver lines (violin plots); it would also be helpful to have feature plots for these genes in a supplement. It is important that some of the data e.g., expression of Slc5a1 and Ffars as well as lack of expression of Piezo2 are tested by e.g., ISH if they deserve a section of text; otherwise this section could be deleted since it is not fundamental to the story.

We now include single-cell transcriptional data for all Cre line markers used later in the study; these genes are highlighted with dashed boxes in the violin plots of Figure 2. We think it is fair to discuss potential roles for genes expressed in our data sets, and reference studies from other labs to support these points.

7. The early focus on lines that are non-specific or not a feature of this paper (including much of Figure 1) and the demonstration of lots of negative expression outside the gut (as well as expression of the Cre-drivers alone) are not really relevant to the story and should be dramatically shortened providing room for more fundamental controls.

We moved some of the data around as suggested, including moving data for the tongue to the main figure (Figure 1D), and some negative images to Figure 1—figure supplement 1. We do think some of the negative images from the intersectional lines are important to illustrate how selective the genetic tools are, and we left some key ones in the main figure.

8. Many strategies are relatively commonplace but are described in great detail, similarly, the nomenclature for the lines needs to be simplified to make this paper more straightforward to read.

Thank you- we have simplified the terminology by replacing "*gene-Cre; Vil1-p2a-FlpO*" with "*gene*^INTER^", which we think will be more straightforward to read.